# Immune cell profiling of the cerebrospinal fluid enables the characterization of the brain metastasis microenvironment

Carlota Rubio-Perez [1,10], Ester Planas-Rigol[1,10], Juan L. Trincado [2,3,10], Ester Bonfill-Teixidor [1], Alexandra Arias[1], Domenica Marchese[2], Catia Moutinho[2], Garazi Serna [1], Leire Pedrosa[4], Raffaella Iurlaro[1], Francisco Martínez-Ricarte[5,6], Laura Escudero [1], Esteban Cordero[5,6], Marta Cicuendez[5,6], Sara Ruiz[2], Genís Parra[2], Paolo Nuciforo [1], Josep Gonzalez[4], Estela Pineda[4], Juan Sahuquillo [5,6], Josep Tabernero [1,6,7], Holger Heyn [2,8,11✉] & Joan Seoane [1,6,7,9,11✉]

Brain metastases are the most common tumor of the brain with a dismal prognosis. A fraction of patients with brain metastasis benefit from treatment with immune checkpoint inhibitors (ICI) and the degree and phenotype of the immune cell infiltration has been used to predict response to ICI. However, the anatomical location of brain lesions limits access to tumor material to characterize the immune phenotype. Here, we characterize immune cells present in brain lesions and matched cerebrospinal fluid (CSF) using single-cell RNA sequencing combined with T cell receptor genotyping. Tumor immune infiltration and specifically CD8$^+$ T cell infiltration can be discerned through the analysis of the CSF. Consistently, identical T cell receptor clonotypes are detected in brain lesions and CSF, confirming cell exchange between these compartments. The analysis of immune cells of the CSF can provide a non-invasive alternative to predict the response to ICI, as well as identify the T cell receptor clonotypes present in brain metastasis.

[1] Vall d Hebron Institute of Oncology (VHIO), Vall d'Hebron University Hospital, 08035 Barcelona, Spain. [2] CNAG-CRG, Centre for Genomic Regulation (CRG), Barcelona Institute of Science and Technology (BIST), Barcelona, Spain. [3] Josep Carreras Leukemia Research Institute and Department of Biomedicine, School of Medicine, University of Barcelona, Barcelona, Spain. [4] Hospital Clinic, University of Barcelona and Institut d'Investigacio Biomedica August Pi i Sunyer (IDIBAPS), Barcelona, Spain. [5] Vall d'Hebron Institut de Recerca (VHIR), Vall d'Hebron University Hospital, 08035 Barcelona, Spain. [6] Universitat Autònoma de Barcelona (UAB), 08193 Cerdanyola del Vallès, Spain. [7] CIBERONC, Barcelona, Spain. [8] Universitat Pompeu Fabra (UPF), Barcelona, Spain. [9] Institució Catalana de Recerca i Estudis Avançats (ICREA), 08010 Barcelona, Spain. [10] These authors contributed equally: Carlota Rubio-Perez, Ester Planas-Rigol, Juan L. Trincado. [11] These authors jointly supervised this work: Holger Heyn, Joan Seoane. ✉email: holger.heyn@cnag.crg.eu; jseoane@vhio.net

Brain metastases (BrM) are the most common tumor of the brain and a devastating complication of cancer with unmet therapeutic needs[1,2]. Immune checkpoint inhibitors (ICIs; e.g anti-PD1, anti-PD-L1, anti-CTLA4) have shown significant clinical benefits in patients suffering from progressive or metastatic solid tumors, including some BrM[3,4]. Still, only a fraction of patients responds to ICI, urging for therapy predictive biomarkers. BrM are genomically and phenotypically different from extra-cranial lesions[5–7] and harbor a unique tumor microenvironment (TME) that includes brain-specific cell types, such as microglia and astrocytes[8]. The composition of the immune TME has been proposed as a predictive marker of response to immunotherapy[9,10]. For example, the degree of inflammation that can be measured by the IFNγ signature[11] and T cell tumor infiltration[12–14] are used to predict response to ICI. Importantly, the analysis of the molecular characteristics that predict clinical responses to ICI rely on the ability to characterize tumor specimens. However, obtaining samples from brain malignancies can be challenging. The anatomical location of brain tumors and related risk of surgical procedures limit the access to tumors. Moreover, the evolving heterogeneity of the TME landscape requires longitudinal analysis and the spatial intra-tumor heterogeneity limits the representativity of single tissue biopsies.

We and others have shown that the cerebrospinal fluid (CSF) can provide fundamental information about the genomic characteristics of brain tumors and hence be used as a relatively non-invasive liquid biopsy[15–18]. Interestingly, studies in non-tumoral diseases, such as multiple sclerosis[19] have shown that CSF leukocytes provide insights into the pathophysiology of the diseases. Thus, we hypothesized that cells in the CSF may reflect the immune TME of BrM. Instead of charting immune cell types from bulk transcriptome analysis (averaging of millions of cells) or using selected markers (flow or mass cytometry); in this work we have decided to use single-cell RNA sequencing (scRNA-seq). This approach provides the resolution required to draw a fine-grained map of the immune TME, comprehensively phenotyping cell types, transient cell states and cancer-specific transcriptomes. High-resolution immunophenotyping through scRNA-seq has been applied to study the immune cell landscapes of several solid primary tumors[20–22], as well as metastatic lesions[7,23].

To comprehensively chart the immune cell landscape of BrM and matched CSF, here we generate high-resolution immune cell maps by applying scRNA-seq combined with T cell receptor (TCR) genotyping. The study is complemented with immunohistochemistry (IHC), flow cytometry (FC) and targeted gene expression analyses. Major cell types, such as T cells, NK cells and tumor-associated macrophages (TAM)/microglia show highly variable frequencies and different phenotypic profiles across patients. Importantly, inflammatory states that can predict ICI response, such as CD8+ T cell tumor infiltration, are recapitulated in the CSF analysis. A continuum of cellular T cell states points to tumor reactivity and clonal expansion, being also detectable in the CSF. Importantly, TCR clonotypes in the CSF match those of the brain lesions, directly linking immune profiles from both compartments.

## Results

### The immune landscape of brain metastases.
To phenotype the inflammatory state of BrM, we analyzed 50 surgical specimens (Fig. 1a, b; Supplementary Data 1; Supplementary Fig. 1a, b; Table 1). BrM specimens were derived from eight distinct primary tumor types, being lung adenocarcinoma (LUAD) the most represented primary lesion (Fig. 1b). In order to assess the degree of inflammation in BrM lesions, we initially used targeted gene expression to determine immune cell infiltration as well as the

IFNγ gene signature enrichment (Supplementary Data 2, 3; Supplementary Fig. 1c, d, g). The samples showed variable inflammatory gene expression profiles allowing their stratification through unsupervised clustering into three immune cell infiltration groups (low, intermediate and high; Fig. 1c). Interestingly, the inflammation level was not associated with the primary tumor site (Supplementary Fig. 1e).

As expected, the IFNγ signature was enriched in the highly immune cell infiltrated group indicating that tumors within this group were inflamed and susceptible to respond to ICI (Fig. 1d). Noteworthy, a high degree of inflammation measured through the IFNγ signature was significantly associated with prolonged overall survival in BrM LUAD patients (Supplementary Fig. 1f). Of note, the tumor with the highest tumor mutational burden (Supplementary Data 4) exhibited a high inflammation (P3) (Supplementary Fig. 1h).

We further performed CD8 IHC in our BrM cohort (Supplementary Data 5). A broad spectrum of CD8+ T cell tumor infiltration was observed ranging from high abundance to almost absence of cells (Fig. 1e). Consistently, we found a significant correlation between cytotoxic lymphocytes (CLym) and IFNγ signature enrichment and CD8+ T cell staining (Fig. 1f). We also performed FC in a set of tumoral samples, based on sample availability (Supplementary Data 6; Supplementary Fig 8, 9a, b). This allowed us to validate the previous results, analyzing the presence and correlation of CD8+ T cells in the tumor (Supplementary Fig. 1i).

### A single-cell atlas of the BrM immune microenvironment.
We applied droplet-based scRNA-seq and sequenced a total of 15,415 high-quality cells from the tumor specimens of nine patients with sufficient material to perform the analysis (Supplementary Figs. 1b, 2a–c). Integrating the single-cell derived transcriptome profiles allowed the clustering of cells and its annotation into major cell types (Supplementary Data 7; Supplementary Fig. 2a, d–f). Cell annotation was done by integrating the genes differentially expressed between clusters, the expression of canonical marker genes and the enrichment of immune cell reference gene sets from the literature (Fig. 2a; Supplementary Figs. 2a, 3, 4, 5, 6). We identified abundant cell types, such as TAM/microglia, CLym (including T and NK cells), and B cells; as well as less abundant cell populations, such as dendritic cells (DC) and neutrophils. T cells were broadly divided into naïve, regulatory (Treg) and cytotoxic T cells (including CD8+ T cells co-clustered with NK cells) (Fig. 2a). All cell types were detected across patients with highly diverse relative proportions underlining the inter-individual heterogeneity in tumor-infiltrating immune cells (Fig. 2b). Interestingly, we identified a cell cluster with an elevated cell cycle signature, indicating active proliferation in T cells and TAMs (Fig. 2a, c).

### BrM immune cell infiltrates are recapitulated in the CSF compartment.
Since the degree of inflammation in BrM can determine response to ICI, we asked whether the analysis of immune cells in the CSF could provide information about the degree of tumor inflammation and provide an alternative to high-risk brain surgeries to guide therapy decisions. Therefore, we performed scRNA and TCR sequencing of six patients with matched BrM-CSF samples, two of them with longitudinal follow-up sampling (Fig. 3a; Supplementary Fig. 1b, 7a–c); analyzing a total of 16 samples. We sequenced 2100 high-quality CSF cells and observed that almost all of them were leukocytes, thereafter named CSF-infiltrating leukocytes (CILs). CILs were readily integrated into a joint single-cell map (Fig. 3b; Supplementary Fig. 7d–f). CSF cells were clustered and annotated following the previously applied rationale identifying all tumor

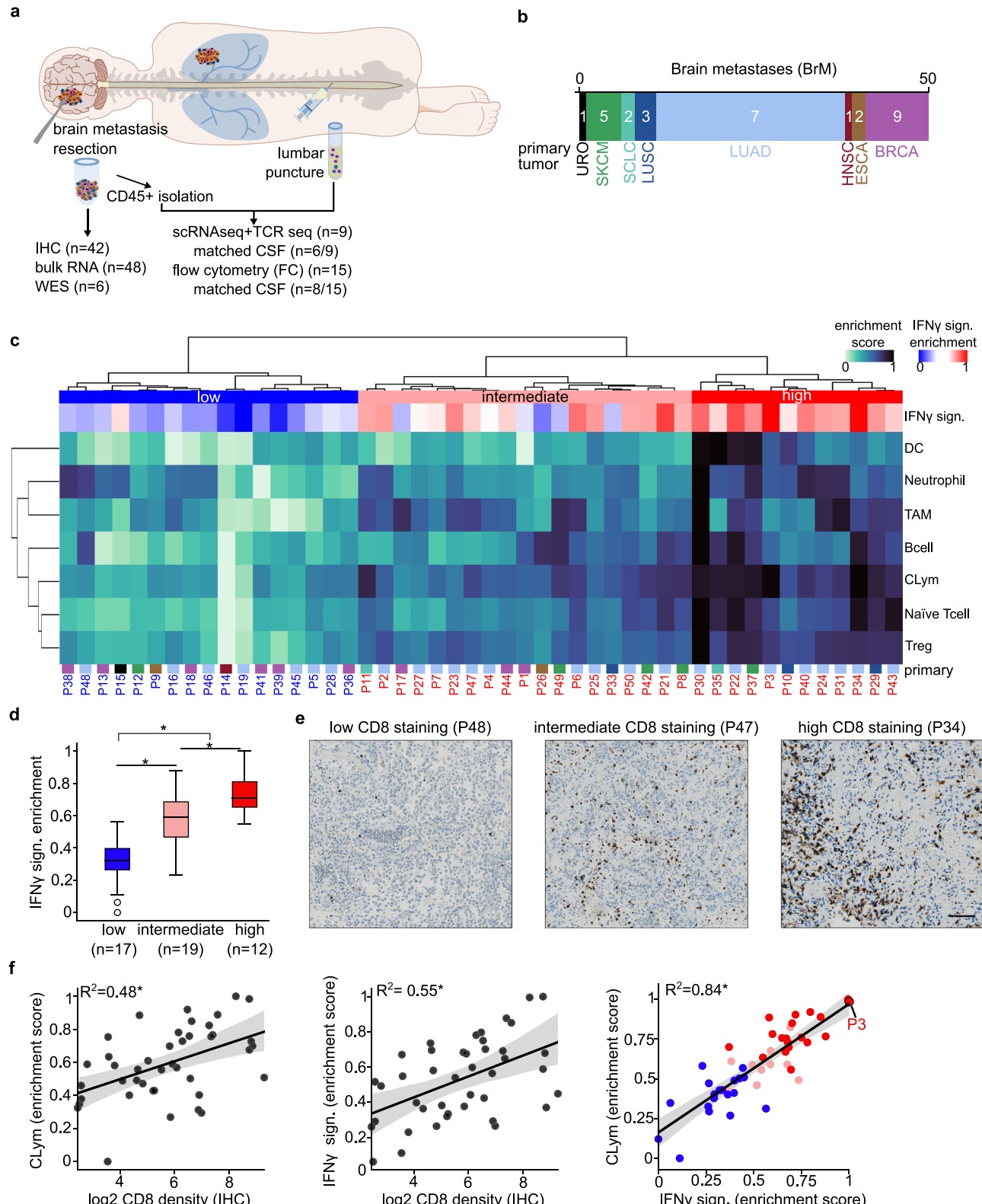

abundant cell types (Fig. 3c, d; Supplementary Data 8, Supplementary Fig. 7d–g). Interestingly, when analyzing the CSF follow-up samples in two patients (P3 and P6), we observed that upon tumor resection, CLym and naïve T cells increased and TAMs decreased in abundance (Supplementary Fig. 7h).

To further investigate and validate the similarity of immune cell type abundance in the CSF and the TME, we analyzed eight patients with matched tumor-CSF samples by FC (Fig. 3a, e; Supplementary Data 6, Supplementary Figs. 1a, b, 8, 9a, b). Importantly, we observed that CD8[+] T and NK cell abundance was similar in the tumor and CSF both by scRNA-seq and FC (Fig. 3e, f). CD4[+] T cells detected by FC were also similarly abundant in the tumor and the CSF (Fig. 3e, f). Noteworthy, we observed a significantly higher CD8[+]/CD4[+] T cell ratio in the tumor compared to the CSF (Supplementary Fig. 9c), in agreement with previous reports[24].

**Fig. 1 Identification of tumor immune infiltration patterns in brain metastases. a** Schematic representation of patient sample collection and the experimental procedures performed. Immunohistochemistry (IHC), whole exome sequencing (WES), cerebrospinal fluid (CSF). **b** Stacked bar plot showing the patient distribution of primary tumor of origin of the BrM (see cancer type acronyms in Supplementary Fig. 1e). **c** Heatmap representing the enrichment scores of seven immune cell types (rows) in 48 BrM patients (columns). IFNγ signature enrichment score has also been represented, in a distinct color scale (see color legend) (see the expression levels by gene in Supplementary Fig. 1g). On top, dendrogram of sample aggregation, according to the enrichment of the seven immune cell types, is shown. Three main clusters have been highlighted (low, intermediate, high); according to the degree of overall infiltration across cell types. On the left, dendrogram of immune cell type aggregation is shown. BrM primary tumor of origin is shown below, following the color scale in **c**. Sample IDs have been colored according to its clustering (blue: low, red: intermediate and high). Cytotoxic lymphocyte (CLym), tumor-associated macrophage (TAM), T cell regulatory (Treg), dendritic cell (DC), signature (sign). **d** Boxplot representing the IFNγ enrichment score distribution across the three clusters of tumor immune infiltration, as shown in **d**. Statistically significant differences are shown with an asterisk (P value < 0.05, T-test). All boxplots indicate median (center line), $25^{th}$ and $75^{th}$ percentiles (bounds of box), and minimum and maximum (whiskers). **e** Representative images of low, intermediate, and high CD8 staining by IHC ($n = 42$ patients). Scale bar, 55 μm. **f** Regression plots representing the Pearson correlation between CLym (left)/IFNγ signature (middle) enrichment score and CD8 IHC cell density ($log_2$ transformed) ($n = 42$ patients). Each dot represents a unique patient sample, Pearson test results are shown ($R^2$ and P value as an asterisk if <0.05), C.I., by performing a multilevel bootstrap, have also been displayed as shaded areas. Right panel represents the Pearson correlation between CLym and IFNγ signature enrichment, samples have been colored according to immune clustering, as shown in **c**, C.I. are also displayed.

**Table 1 Summary of the main clinical features of the cohort of study.**

| Characteristic | BrM ($n = 50$) |
| --- | --- |
| Ratio female/male | 1.04 |
| Median age at diagnosis—years (±SD) | 58 (13) |
| Median overall survival—months (±SD) | 38 (39) |
| Ratio dead/alive | 1.3 |
| Pre-biopsy treated with CT or any other therapy—number (%) | 33 (66) |

*BrM* brain metastases, *CT* chemotherapy, *SD* standard deviation.

**TCR profiling in BrM and the CSF**. TCR sequencing provided 2082 sequences (alpha and beta regions, Supplementary Data 9; Supplementary Fig. 10a, b) for 1729 cells. TCR sequences were further divided into singleton (1813 sequences) or expanded (269 sequences) based on the number of cells supporting each sequence (referred to as TCR clonotype) and mapped onto the scRNA-seq immune cell map (Fig. 4a). Expanded TCR clonotypes were significantly enriched in cytotoxic T cell populations including the subset of proliferating cytotoxic T cells; while singleton clonotypes were significantly enriched in naïve T cells and Tregs (Fig. 4b, c). In line, T cells with clonal expansion expressed *GZMA/B*, *PRF1*, *INFG*, and *MIK67*, indicating their tumor-reactive and cytotoxic state (Fig. 4d). Pseudotime trajectory inference ordered T cells in a continuum, originating from naïve and concluding with reactive/proliferating cytotoxic states and Tregs (Fig. 4e). Accordingly, the degree of TCR clonal expansion increased gradually in the pseudotime trajectory suggesting concerted activation and proliferation programs (Fig. 4e). For individual tumors, we observed a wide range of TCR clonotype composition, being P3 the patient bearing a tumor with the most heterogeneous clonal profile (Fig. 4f).

Importantly, we detected identical TCR sequences between the CSF and the metastatic lesions in four out of six patients, confirming the direct interaction between both compartments (Supplementary Fig. 10c, d). Moreover, identical TCR clones were detectable at multiple sampling time points (Fig. 4g, h; Supplementary Fig. 10e). The longitudinal follow-up further suggested selected TCR clones to be maintained over the course of the treatment, while others could not be traced over multiple sampling time points (Fig. 4h, i; Supplementary Fig. 10f). Together, this indicates the possibility to monitor the BrM immune and TME T cell subclonal evolution during cancer progression through the analysis of CILs.

## Discussion

The detection of varying immune compositions and inflammation states could determine the response to immunotherapy and lays the ground for patient selection and stratification in the context of ICI therapies. Single-cell profiling allowed us to chart major cell types and transient tumor-specific states present in the BrM TME. Most interestingly, we observed that the phenotype of CLym in the CSF recapitulated the one observed in the tumor, providing a relatively non-invasive tool to characterize and assess the degree of inflammation in brain lesions that in turn could be used to predict response to ICIs.

The nature of the TME can dictate the biology of tumors and determine the tumor sensitivity to immune therapies. Importantly, the brain TME has unique characteristics[6,7]. The singularity of the brain TME and the divergence of the brain lesions compared to extra-cranial lesions emphasize the requirement to characterize and monitor the brain TME in each patient in order to evaluate whether immune therapies could be beneficial. However, the anatomical location of the tumor limits its accessibility and thus the characterization of the brain TME. Our observation that the CSF can recapitulate the immune landscape of the brain lesion indicates the analysis of the CSF to provide critical information about the brain TME in a relatively non-invasive manner, avoiding intracranial surgeries.

In the present work, we did not aim to perform a detailed characterization of the immune cells but focused on evaluating the levels of CLym, information relevant to determine ICI clinical responses. Future studies are warranted to thoroughly characterize in detail each of the immune cell types present in the CSF and compare them to the tumor lesions.

Interestingly, the CSF has been previously analyzed to characterize the pattern of leukocytes in other diseases, such multiple sclerosis[19]. Here, scRNA-seq revealed a similar immune landscape to the one we observed in BrM. This further reinforces the relevance of the CSF as a source to characterize the immune system in brain diseases.

Noteworthy, we identified matching TCR clonotypes of cytotoxic T cells in the CSF and the brain tumor. This demonstrates the interdependence of both compartments and, importantly, shows that tumor-reactive T cells present in the CSF could be used in a CSF-guided TCR cloning strategy for cell therapy or vaccines.

Together, our results show that the CSF immune cell profile can facilitate the characterization of the immune TME in brain metastatic lesions and longitudinally monitor the evolution of the cancer immune response.

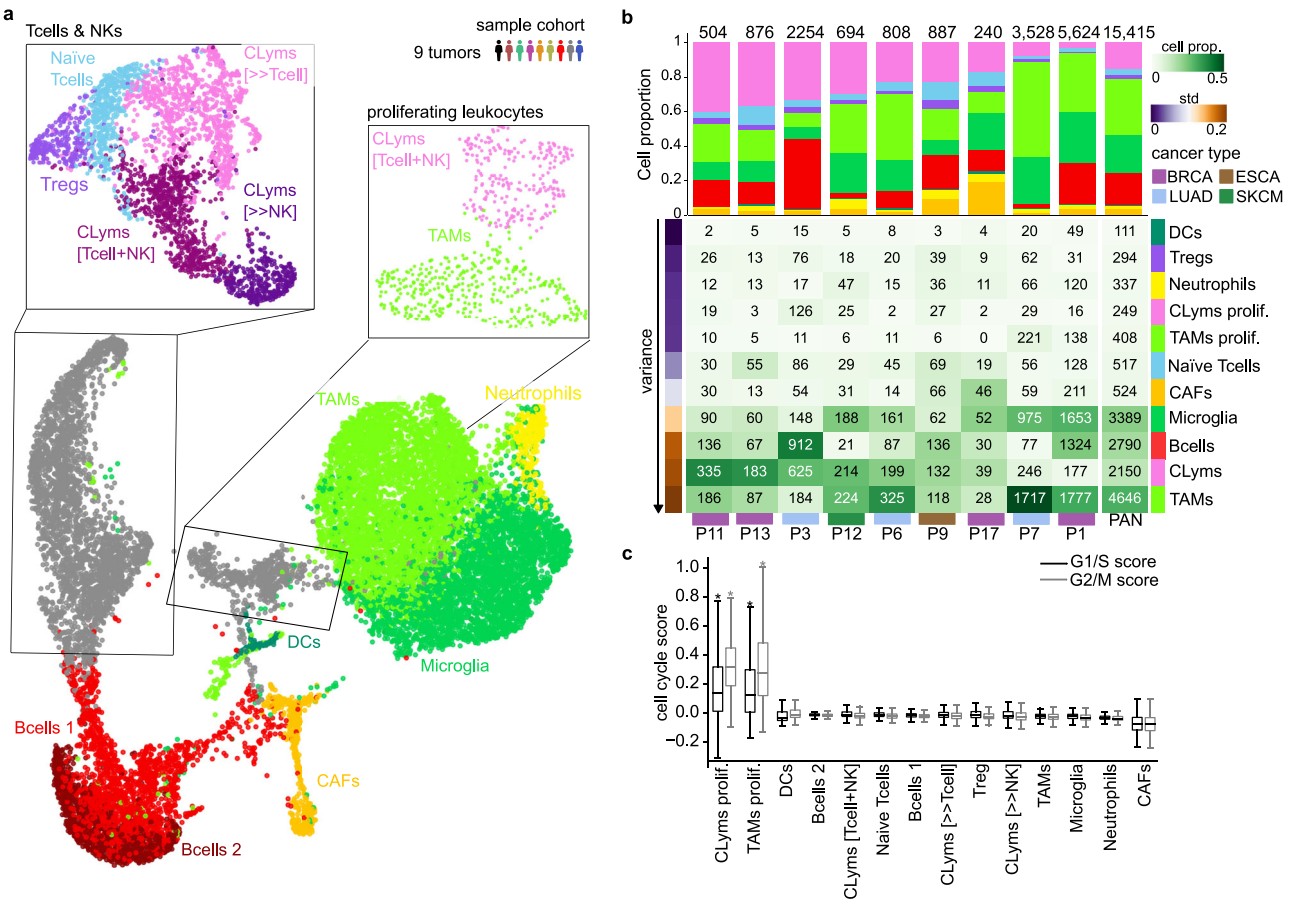

**Fig. 2 High-resolution map of immune infiltration of brain metastases. a** UMAP projection of tumor-infiltrating cells colored according to cell lineages ($n = 9$ patients, $n = 9$ tumor samples, $n = 15{,}415$ cells). On top right, a diagram representation of the sample cohort considered for all figure panels is shown. A detailed UMAP projection is shown for T cell and NK and proliferating leukocyte cell clusters. Cytotoxic lymphocyte (CLym), cancer-associated fibroblast (CAF), tumor-associated macrophage (TAM), T cell regulatory (Treg), dendritic (DC), natural killer (NK). **b** Heatmap showing the relative abundance of tumor-infiltrating cell type, measured as the proportion of cells of the cell type vs. the total of sequenced cells, across tumor samples in the UMAP projection. Some cell types have been aggregated: B cells 1 and 2 into B cells and CLym [T cell + NK], [» T cell] and [»NK] into CLym. Rows represent cell types, sorted by variance (represented on the left, standard deviation from low to high). Columns represent patients, sorted according to the relative abundance of CLym; the primary tumor of origin is shown below. The last column represents all patients together (pancancer, PAN). Top panel shows a stacked bar plot representing the relative abundance of each cell type. Color legend is shown in the cell type labels of the heatmap. **c** Boxplot representing G1/S and G2/M scores across cell clusters in **a**. Each boxplot represents a cell distribution, the total of cells represented in each boxplot is found in **b** PAN column. Statistically significant differences where fold change is >0.1 are shown with an asterisk ($P$ value < 0.05, Mann–Whitney $U$ test). All boxplots indicate median (center line), 25th and 75th percentiles (bounds of box), and minimum and maximum (whiskers). Proliferating (Prolif).

## Methods

**Patients.** Human BrM samples ($n = 50$ patients) were obtained from the Vall d'Hebron University Hospital and Clinic Hospital. The protocol to obtain samples was approved by the Hospital IRB (PR(AG)478/2017) and informed consent was obtained for all patients. Electronic health records of the patients were expert reviewed and a database of clinical annotations was built (Supplementary Data 1).

Due to sample availability, not all samples had sufficient material to be analyzed with all the techniques described in the manuscript (Supplementary Data 1). Hence, distinct sample cohorts were used in the different analyses performed (Supplementary Fig. 1a, b).

**Cell sorting for single-cell RNA and TCR sequencing.** We collected 19 samples from 9 patients suffering from BrM of distinct tissues of origin. For 6/9 patients a matched CSF sample, prior to surgery (named CSF $t_0$) and in two cases after surgery (CSF $t_1$; +1 month from surgery) and after treatment (CSF $t_2$; +3 months) were collected (Supplementary Data 1; Supplementary Fig. 1b). Thus, a total of ten CSF samples were analyzed. After resection, tumor specimens were enzymatically digested (human tumor dissociation kit, Miltenyi) and CD45+ cells were isolated using human CD45 TIL microbeads (Miltenyi) and stored in PBS containing 0,005% BSA. CD45+ isolation purity was between 80 and 90% depending on the sample. CSF was extracted from the patient through a lumbar puncture, as a part of its health care. From it, we separated 3 mL which were centrifuged at $400 \times g$ for 10 min to obtain the cells which were stored in PBS + 0.005% BSA. No CD45+ isolation was performed in the CSF samples.

When samples could not be immediately processed, cold methanol was carefully added to the samples and kept at −80 °C overnight until processed. Other samples were cryopreserved with inactivated FBS with 10% of DMSO.

**Single-cell RNA and TCR sequencing.** The 19 samples were loaded into the 10× Genomics Chromium Controller for droplet-encapsulation. Single-cell gene expression and TCR clonotypes (TCR) were produced using the Chromium Single-Cell 5′ Library, following the manufacturer's instructions, and sequenced on an Illumina NovaSeq 6000. One sample (P6 CSF $t_0$) failed the quality control of TCR enrichment PCR and the final TCR library did not show any detectable amplification. This could be explained by a low cell number in P6 CSF $t_0$ sample, below the limits of sensitivity of 10X Single-Cell V(D)J kit.

**Single-cell RNA data analysis.** scRNA sequenced reads were aligned and quantified through CellRanger Single-Cell Software Suite (v3.1.0) to Homo Sapiens Hg38 reference genome. Next, we performed a quality control and filtering of the cells. In tumor samples, we filtered out cells: (1) with less than 100 genes detected (to filter cells with low RNA integrity and erythrocytes), (2) with more than 10% of expressed mitochondrial genes (to filter degraded, broken cells) and (3) with more than 2500 genes expressed (to filter cell doublets). In CSF specimens we filtered out cells: (1) with a given low number of expressed genes (raging between 70 and 100), determined based on the maximization of erythrocyte filtering; (2) with more than 20% of expressed mitochondrial (MT) genes and (3) with cells with more than 2500 expressed genes (see the whole in silico analysis flow in Supplementary Fig. 2a).

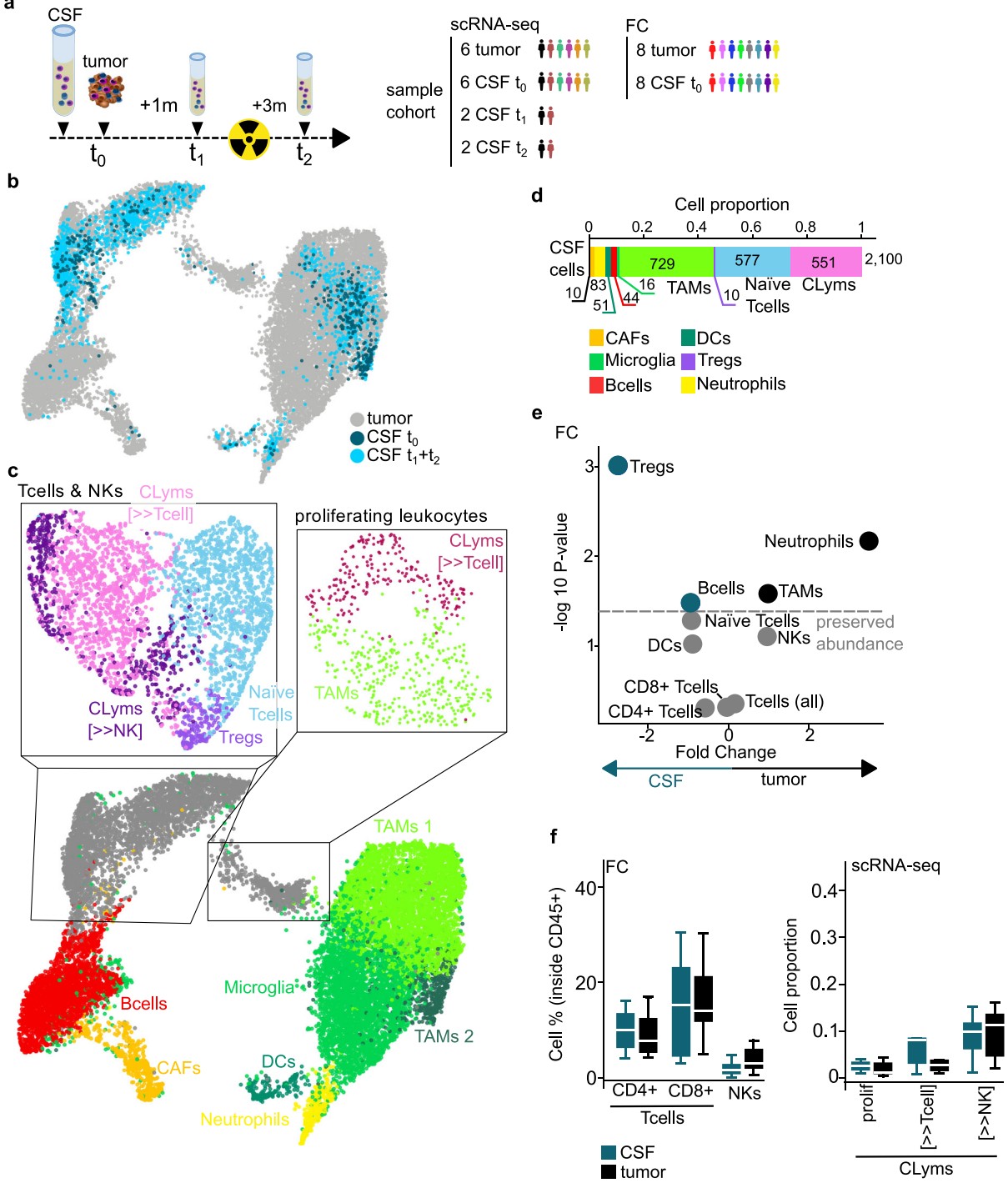

The values of % of mitochondrial genes and the number of detected genes per cell are available in Supplementary Data 7, 8.

The following downstream analyses were performed through Seurat library (v3.0.2) for R 3.4.4[25]. To remove batch effect by sample preservation technique (methanol vs. fresh vs. cryopreservation) samples were integrated following Seurat *anchor integration* method. It can be observed in Supplementary Fig. 2d–f how the batch effect by preservation technique could be properly adjusted, and that there was no batch by sample either.

Next, we sought to identify cell clusters, through Seurat. Variable genes were called at the cohort level through *vst* method, with default parameters. Gene UMIs were z-scaled and used as input for principal component analysis. Standard deviations of the principal components were calculated to determine the dimensions to be considered in the cell clustering. Cell clusters were identified using the Louvain community clustering algorithm with a variable number of

dimensions depending on principal component variability (SD ≈ 2). Cluster resolution including all cells in the cohort was set to 0.4. The same clusters were used to generate the UMAP projections. In addition, we assigned scores for G1/S and G2/M cell cycle phases across clusters based on previously defined gene sets[20].

To better differentiate T cell subpopulations we did a re-clustering strategy of cell clusters with: >10% cells with average (*CD3E/CD3D*) expression >0. To avoid gene variability dilution, we did not adjust for cell cycle but re-clustered alone those clusters with a strong cell cycle effect. We manually checked for the existence of cells co-expressing T cell (e.g., *CD3D/CD3E*) gene markers and myeloid markers (e.g., *CD68*) to ensure that doublet cell removal was effective.

Cell annotation was performed based on a three-step strategy, which provided distinct layers of information to improve the accuracy of the annotation. (1) We built a set of in-house expert-curated gene markers, for each cell cluster we checked the % of cells expressing each of them along with the expression range

**Fig. 3 Immune cell type identification in the CSF of BrM patients. a** Left, schematic representation of BrM tumor resection and cerebrospinal fluid (CSF) collection at different time points: $t_0$, initial tumor resection and CSF collection at the day of surgery and prior to it; $t_1$, CSF collection 1 month (1m) after tumor resection; $t_2$, CSF collection after 3 months (3m) of tumor resection. Holocranial radiotherapy was administered between $t_1$ and $t_2$. Right, diagram representation of the sample cohort considered for all figure panels related to scRNA-seq ($n = 6$ patients, $n = 16$ samples) (**b–d**, **f**); and flow cytometry (FC) ($n = 8$ patients, $n = 16$ samples) (**e**, **f**). If not specified all CSF samples have been included in the downstream analyses. **b** UMAP projection of tumor-infiltrating and CSF cells, colored according to sample type $n = 15,895$ cells. **c** UMAP projection of tumor-infiltrating ($n = 13,795$) and CSF cells ($n = 2100$) colored according to cell lineages. A detailed UMAP projection is shown for T cell and NK and proliferating leukocyte cell clusters. Cytotoxic lymphocyte (CLym), cancer-associated fibroblast (CAF), tumor-associated macrophage (TAM), T cell regulatory (Treg), dendritic cell (DC), natural killer (NK). **d** Stacked bar plot representing the relative abundance of all cells identified by scRNA-seq in the CSF. Absolute numbers of identified cells per cell type are shown. Cell types are colored as shown in color legend. **e** Volcano plot comparing cell type relative abundance (inside the fraction of CD45$^+$ cells) in the tumor vs. the CSF ($n = 8$ patients), according to FC experiments. x-axis represents the $-\log_2$ fold change and y-axis the $-\log_{10} P$ value according to Mann–Whitney $U$ test. Each dot represents a cell type: non-significant differences are colored gray, cell types significantly overrepresented in the tumor are colored black and cell types significantly overrepresented in the CSF are colored blue ($P$ value < 0.05). **f** Paired boxplot representing the relative abundance of lymphocyte types in the tumor (black) vs. the CSF (blue). Left panel (FC sample cohort), distribution of the percentage of CD4$^+$ and CD8$^+$ T cells and NK cells identified in the tumor and CSF, inside the fraction of CD45$^+$ cells. Right panel (scRNA-seq sample cohort), distribution of the relative abundance of cytotoxic lymphocytes in the tumor and CSF. Statistical significance in both panels was calculated with Mann–Whitney $U$ test, non-significant results were found ($P$ value > 0.05). All boxplots indicate median (center line), 25$^{th}$ and 75$^{th}$ percentiles (bounds of box), and minimum and maximum (whiskers).

(Supplementary Fig. 3b, d; 4a; 5a, b, e; 6a, b). We also performed differential gene expression (DE) across cell clusters (performed through Seurat *FindAllMarkers* method) and considered directly for cell annotation the top-gene DE between clusters (Supplementary Figs. 3a, 5c, 6c). We also performed a pre-ranked Gene Set Enrichment analysis (GSEA)[26] based on the fold change of the DE analysis of high-quality immune cell gene sets gathered from the literature (through Python 3.1 *gseapy* package: http://gseapy.rtfd.io/) (Supplementary Figs. 3c, 5d). To be more restrictive we considered all positive enrichments with a false discovery rate (FDR) <0.25 (according to GSEA documentation, enrichments with an FDR <0.25 are significant).

We did two additional analyses to empower the TAM vs. Microglia cell annotation (Supplementary Fig. 4b, c): (1) we identified the genes DE between these two cell groups, did a manually curated revision and identified cell lineage markers; (2) we downloaded the *Macrophage* and *Microglia* gene sets defined by and performed a GSEA (following the same methodology above) on the ranked list of DE genes between TAM and Microglia cell groups, previously identified[27].

Trajectory analysis was performed with the Monocle package (v2.16.0)[28]. The highly variable genes obtained for the integration of the data via Seurat were used for pseudotime ordering. Dimensionality reduction was applied with the DDRTree option.

**TCR data analysis**. TCR enriched libraries were mapped using CellRanger Single-Cell Software Suite (v3.1.0) with *vdj* option to the GRCh38 VDJ pre-built reference provided by 10x (v3.1.0). TCR clonotypes were obtained from cells identified as true cells by CellRanger, containing full-length recombinant sequences and productive CDR3 chains. Cells sharing the exact CDR3 amino acid region were considered as clones. Exact same alpha or beta chain sequence was considered a clonotype.

In CSF samples where multiple sample time points were available ($t_0$, $t_1$, $t_2$) a clonal evolution analysis was performed, with Fish Plot R package[29], TCR clones were treated as independent samples.

**Flow cytometry**. We collected 16 samples from 8 patients with BrM of distinct tissues of origin with a matched CSF $t_0$ sample. We pooled these together with seven tumoral samples, which were already analyzed by scRNA and TCR sequencing (Supplementary Data 1; Supplementary Fig. 1a, b), as the FC sample cohort.

After resection, tumor specimens were enzymatically digested for 13/15 (human tumor dissociation kit, Miltenyi) and CD45$^+$ cells were isolated using CD45 TIL microbeads (Miltenyi). For the other two samples, no CD45$^+$ isolation was performed (Supplementary Data 6, T: tumor, non CD45$^+$ isolation; CD45: CD45$^+$ isolation from the tissue). Cells from the CSF were obtained by centrifugation at $400 \times g$ 10 min. No CD45$^+$ isolation was performed.

For immune cell characterization of tumor and CSF samples, anti-human antibodies against CD11b-PE (clone D12, 34755, 1:100, BD), CD45-BV510 (clone HI30, 304036, 1:100, Biolegend), CD3-BV650 (clone, SK7, 563999, 1:100, BD Bioscience), CD8-APC-H7 (clone SK1, 560179, 1:100, BD Bioscience), CD4-BV786 (clone OKT4, 317441, 1:100, Biolegend), CD66b-FITC (clone G10F5, 555724, 1:100, BD Bioscience), CD56-APC (clone Tuly56, 17-0566-42, 1:100, Thermofisher), CD19-AlexaF700 (clone HIB19, 302225, 1:100, Biolegend), and CD11c-BV605 (Clone 3.9, 301636, 1:100, Biolegend) were used. Samples were previously incubated with Fixable Viability Stain 620 (PE-CF594) (564996, BD BIoscience) to determine viability. Samples consisting on CD45$^+$ isolated cells from digested tumors with a spike of peripheral blood mononuclear cells (PBMCs) were used to establish leukocyte populations (named as "spike"). 10$^6$ PBMCs were added into the single staining and FMOs samples in order to establish the leukocyte populations among the samples. Isotype control of anti-human CD45-BV510 (mouse IgG1k-BV510, 400171, 1:100, Biolegend) was also used as a control (Supplementary Fig. 8a).

PBMCs were obtained from donors' buffy coats. Centrifuge density separation using Lymphosep (Biowest) was performed to obtain the PBMCs.

Immune cell populations (inside alive cells) were determined by the following cell markers as shown in Supplementary Fig. 8b: total leukocytes (CD45$^+$), T cells (CD45$^+$,CD3$^+$), CD8$^+$ T cells (CD45$^+$,CD3$^+$,CD8$^+$), CD4$^+$ T cells (CD45$^+$,CD3$^+$, CD4$^+$), TAMs (CD45$^+$, CD3$^-$, CD11b$^+$/CD66b$^-$), Neutrophils (CD45$^+$, CD3$^-$, CD11b$^-$/CD66b$^+$), NK (CD45$^+$, CD3$^-$, CD56$^+$), B cells (CD45$^+$, CD3$^-$, CD19$^+$), DCs (CD45$^+$, CD3$^-$, CD11c$^+$).

For T cell characterization of tumor and CSF samples, anti-human antibodies against CD45-PE-Cy5 (clone HI30, 15-0459-42, 1:100, Thermofisher), CD3-BV650 (clone, SK7, 563999, 1:100, BD Bioscience), CD45RA-PE-cy7 (clone HI100, 304126, 1:100, Biolegend), CCR7-FITC (clone G043H7, 353216, 1:100, Biolegend) and FOXP3-Alexa647 (clone 259/c7, 560045, 1:50, BD Bioscience) were used. Samples were previously incubated with LIVE/DEAD fixable yellow dead stain kit (L34959, 1:2000, ThermoFisher Scientific) to determine viability. As a positive control, activated PBMCs obtained from buffy coats from donors were used. To activate PBMCs, non-treated 24-well plate was coated with 5 µg/ml/per well of anti-human CD3 (in PBS) (clone OKT3 monoclonal antibody, 16-0037-85, ThermoFisher Scientific). After 2 h of incubation at 37 °C, the plate was washed and 10$^6$ PBMCs with 2 µg/ml of soluble anti-CD28 (clone CD28.2 monoclonal antibody, 16-0289-85, ThermoFisher Scientific) in RPMI 1640 (supplemented with Glutamine and inactivated FBS) were added per well and activated for 72 h. 10$^6$ activated PBMCs ("activated PBMCs") were added into the single staining and FMOs samples in order to establish the leukocyte populations among the samples. Isotype control of anti-human CD45-PeCy5 (mouse IgG1k-PeCy5, 14-4714-81, 1:100, eBioscience) was also used as a control (Supplementary Fig. 9a).

T cell subpopulations (inside alive cells) were determined with the following cell markers as shown in Supplementary Fig. 9b: total leukocytes (CD45$^+$), T cells (CD45$^+$,CD3$^+$), Regulatory T cells (Treg) (CD45$^+$, FOXP3$^+$/CD3$^+$), Naïve T cells (T naïve) (CD45$^+$, CD3$^+$, CD45RA$^+$/CCR7$^+$).

Samples were acquired on a BD FACSCELESTA (immune cell characterization panel) or BD LSRFortessa™ cell analyzer (BD Biosciences) (T cell characterization panel) and data were analyzed using FlowJo software.

**Immunohistochemistry**. IHC antibody (790–4460 Roche-Ventana) against anti-CD8 (clone SP57, 790–4460) was used to stain slides from paraffin-embedded tissues. Slides were deparaffinized and hydrated using Discovery Ultra IHC/ISH Platform (Roche). Antigen retrieval was performed using Discovery CC1 buffer for 64 min at 95 °C. Subsequent incubation of 8 min with CM inhibitor was used for peroxidase blockade. Incubation of primary antibody (anti-human CD8) was performed for 20 min at 36 °C with subsequent 8 min incubation with UltraMap anti-Rabbit antibody (HRP). As a detection system, CM ChromoMap was used according to the manufacturer's instructions, followed by counterstaining with hematoxylin II (8 s), dehydration and mounting.

IHC of CD8 and corresponding quantifications were performed in 42 different patients with BrM (Supplementary Data 1, 5). For CD8 IHC quantifications, Visiopharm software was used. To calculate cell density the total number of positive cells was divided by the tumoral area (positive cells/mm$^2$). Cell density was log$_2$-transformed.

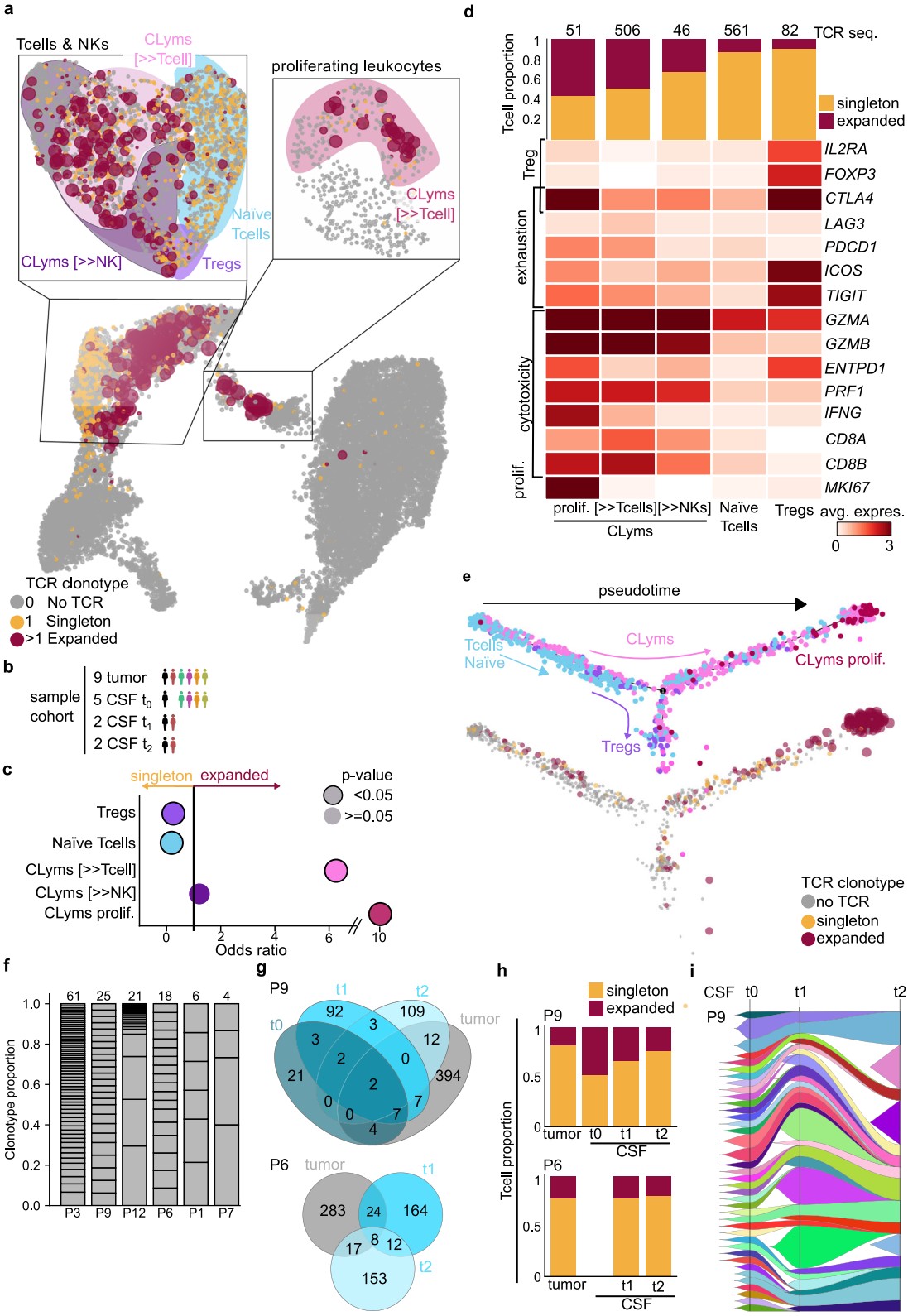

**Targeted RNA profiling**. RNA from 48 BrM samples was extracted and expression levels for 730 immune-related and 40 additional housekeeping genes were obtained, through PanCancer Immune Panel v1 (Nanostring®)[30]. A minimum of 250 ng of total RNA was used as input of Nanostring Pancancer Immune Profiling panel. Expression levels for 730 immune-related and 40 additional housekeeping genes were obtained.

Samples with a geometric mean expression lower than 50 counts in the housekeeping genes were discarded due to low quality (Supplementary Fig. 1c).

For some samples, with large tissue availability ($n = 7$), duplicates were included (named as Px_2). In these cases, the sample with the highest geometric mean expression of HKs genes was included and the sample was re-named as Px (Supplementary Fig. 1c).

Next, raw data processing was performed with NSolver® analysis software. Expression values were adjusted for background noise thresholding by the maximum number of counts of the negative controls. Data normalization was performed using the geometric mean of the counts of the positive controls and the

**Fig. 4 T cell profiling in the tumor and CSF of BrM patients. a** UMAP projection of tumor-infiltrating and cerebrospinal fluid (CSF) cells ($n = 6$ patients, $n = 16$ samples, $n = 15,895$ cells), colored according to T cell receptor (TCR) clonotype (expanded vs. singleton). Cell size represents the degree of expansion of TCR clonotypes. A detailed UMAP projection is shown for T cell and NK and proliferating leukocyte cell clusters. Cell type clusters have been delimited and background colored (as Fig. 3c). Cytotoxic lymphocyte (CLym), cancer-associated fibroblast (CAF), tumor-associated macrophage (TAM), T cell regulatory (Treg), dendritic cell (DC), natural killer (NK). **b** Diagram representation of the sample cohort considered for all downstream figure panels ($n = 6$ patients, $n = 15$ samples). **c** Dot plot representing the enrichment (odds ratio) of expanded vs singleton clonotypes across T cell and NK clusters, by Fisher exact test. **d** Heatmap representing the average gene expression of functional T cell markers (rows) across cell T cell and NK clusters as labeled in **a** (columns); sorted by the fraction of cells with expanded TCR clonotypes. Top panel shows a stacked bar plot representing the fraction of T cells with expanded and singleton clonotypes across each cell cluster. **e** Pseudotime projection of T cells, filtered by cells where TCR was sequenced. Top panel shows T cells colored by cell type and bottom panel by TCR clonotype expanded vs. singleton. Cell size in the bottom panel represents the degree of clonotype expansion. **f** Stacked bar plot representing the abundance of each expanded TCR clonotype, in tumor samples, across patients. **g** Venn diagram displaying the overlap of identical TCR sequences across the distinct sample types (tumor, CSF $t_0$, CSF $t_1$, and CSF $t_2$) in two patients (P6 and P9). **h** Bar plot representing the fraction of T cells with expanded and singleton clonotypes across the sample types and patients shown in **f**. **i** Fish plot representing the clonal evolution of CSF TCR sequences in three distinct time points ($t_0$, $t_1$, and $t_2$) in patient P9.

---

housekeeping genes. No batch effect was detected, see Supplementary Fig. 1d. Expression values were log₂- and z-score transformed across samples. We calculated the enrichment scores of the distinct gene sets of interest: seven immune population gene sets built with reference markers and IFNγ signature[11]; according to the following formula:

$$\frac{patientZ_i - min(patientsZ)}{max(patientsZ) - min(patientsZ)} \quad (1)$$

Equation 1. Enrichment score. $\mu$ refers to average $\sigma$ to standard deviation and patientsZ is defined as in Eq. 2.

$$\frac{\sum_{n-i}^{i} \frac{gene_i - \mu(gene\ across\ patients)}{\sigma(gene\ across\ patients)}}{n(signature\ genes)} \quad (2)$$

Equation 2. patientsZ calculation. where $n$ refers to number, $\mu$ to average $\sigma$ to standard deviation.

Hierarchical agglomerative clustering of the tumor samples was performed using the Ward linkage method, through *sklearn* Python 3.1 package. The number of clusters was set to three.

**Whole exome sequencing**. We performed whole exome DNA sequencing (WES) of the tumors of six patients in both the tumor and germline DNA. DNA from 12 samples were extracted through QIAamp DNA mini Kit (Qiagen). Nimblegen SeqCap EZ MedExome + mtDNA (Roche, 47 Mb) was used to perform whole exome enrichment. The libraries were sequenced on HiSeq2500 (Illumina) in paired-end mode with a read length of $2 \times 100$ bp. Each sample was sequenced in a fraction of a sequencing v4 flow cell lane, following the manufacturer's protocol. Image analysis, base calling and quality scoring of the run were processed using the manufacturer's software real-time analysis (1.18.66.3) and followed by generation of FASTQ sequence files.

WES reads were mapped to Hg19 using the GEM3 toolkit[31]. Alignment files (BAM format), containing only properly paired, uniquely mapping, reads; were processed using Picard (v.1.110) (https://broadinstitute.github.io/picard/) to add read groups and remove duplicates. The resulting BAM files were processed using SAMtools (v. 1.2)[32,33] and the GATK (v. 3.2.0). Somatic tumor variants were called by Mutect2[34] and Strelka[35] doing a matched normal analysis. Only those mutations reported by both callers were considered for further analyses.

The biological relevance of all mutations was annotated through Cancer Genome Interpreter (CGI; v1907)[36], which was run considering the primary tumor of origin as input cancer type. The mutational tumor burden was calculated considering as numerator the number of non-synonymous mutations (as predicted by CGI) and the library size (47 Mb) as denominator.

**Statistical analyses**. All statistical analyses were performed with Python 3.7 SciPy library[37], including the following calculations: Entropy, Spearman correlation, Pearson correlation, T-test, Mann–Whitney $U$ test, Wilcoxon test and Fisher exact test. Survival analysis, including Kaplan–Meier and log-rank tests, were performed through Python 3.7 lifelines package. All statistical tests have been performed as two-sided. Exact $p$ values of statistical analyses can be found in Source Data.

## Data availability
All sequence data, including scRNA, TCR, and WES have been deposited in the European Genome Archive (EGA) under accession code EGAS00001004751. Nanostring data have been deposited in Gene Expression Omnibus (GEO) and is under accession code GSE159407. Source data are provided with this paper.

## Code availability
The R and Python codes, and software dependencies, to reproduce all paper figures are available in bitbucket (https://bitbucket.org/carlotarp1/brainmets_csf).

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

## Acknowledgements

The authors would like to thank the patients at the Vall d'Hebron Hospital and Clinic Hospital that were enrolled in the study and their families. The study was undertaken with the support of the Fundación Asociación Española contra el Cáncer (AECC) (EDM), FERO (EDM), Ramón Areces Foundation, Cellex Foundation, BBVA (CAIMI), the ISCIII, FIS (PI16/01278), the Ministerio de Ciencia, Innovación y Universidades (SAF2017-89109-P; AEI/FEDER, UE), the Juan de la Cierva formación fellowship (C.R.-P. and L.E.), Sara Borrell fellowship (E.P-R.). We acknowledge support of the Spanish Ministry of Science and Innovation to the EMBL partnership, the Centro de Excelencia Severo Ochoa and the CERCA Program/Generalitat de Catalunya. We also acknowledge the support of the Spanish Ministry of Science and Innovation through the Instituto de Salud Carlos III, the Generalitat de Catalunya through Departament de Salut and Departament d'Empresa i Coneixement, and the Co-financing by the Spanish Ministry of Ministry of Science and Innovation with funds from the European Regional Development Fund (ERDF) corresponding to the 2014–2020 Smart Growth Operating Program.

## Author contributions

J.S. and H.H. supervised the whole project. J.S. conceived the project. C. R-P., E. P-R., and JL.T. performed most of the experiments, including wet lab and bioinformatic analyses. J.S., H.H., C. R-P., E. P-R. and JL.T. interpreted the results. J.S and H.H. wrote the manuscript in collaboration with C. R-P. and E. P-R. E. B-T., A.A., R. I., G.S., D. M., C. M., and S. R. contributed to the wet lab experiments. G. P. contributed to the bioinformatic analyses. F. M-R., L.E., E. C., L. P., J. G., E.P, J.Sa., P. N., and J.T. coordinated patient clinical data and samples, as well as provided valuable critical discussion.

## Competing interests

J.S. is co-founder of Mosaic Biomedicals and has ownership interests from Mosaic Biomedicals and Northern Biologics. J.S. received grant/research support from Mosaic Biomedicals, Northern Biologics, Roche/Glycart, and Hoffmann la Roche. J.S. declares scientific consultancy role for Merck Serono, GSK, Eli Lilly. J.T. declares scientific consultancy role for Array Biopharma, AstraZeneca, Bayer, BeiGene, Boehringer Ingelheim, Chugai, Genentech Inc., Genmab A/S, Halozyme, Imugene Limited, Inflection Biosciences Limited, Ipsen, Kura Oncology, Lilly, MSD, Menarini, Merck Serono, Merrimack, Merus, Molecular Partners, Novartis, Peptomyc, Pfizer, Pharmacyclics, ProteoDesign SL, Rafael Pharmaceuticals, F. Hoffmann–La Roche Ltd., Sanofi, SeaGen, Seattle Genetics, Servier, Symphogen, Taiho, VCN Biosciences, Biocartis, Foundation Medicine, HalioDX SAS, and Roche Diagnostics. E.P. declares scientific consultancy role for Celgene. P.N. declares scientific consultancy role for Bayer, Novartis and Merck Sharp and Dohme. The other authors declare no competing interests.
