## [Peer Review File · Nature Communications]

REVIEWER COMMENTS

Reviewer #1 (Remarks to the Author):

This is an interesting study by Rubio-Perez and colleagues about studying cell-type specific diversity and transcriptomic changes in the CSF of patients with brain metastasis tumor formation. The group addressed the question whether the cellular/molecular diversity in the CSF does reflect changes in the actual tumor microenvironment. They applied scRNA-seq in combination with TCR genotyping, and found conserved TCR clonotypes between matched CSF and brain tumor samples. Also, by characterizing the immune cell signatures in the CSF, predictions about responses towards immune checkpoint inhibition were made. In summary, this manuscript follows-up on the idea that by analyzing a "liquid biopsy" compartment such as the CSF, it might be possible to study the microenvironment of solid tumors and predict response patterns to specific immune cell therapies. Overall, this is a well conducted and elegant study, however, several points remain that need to be addressed. Also, it would be interesting to see if further differences between the variable primary tumors can be further demonstrated with respect to immune subtypes and TCR subclones, which would be critical when tailoring immune therapies.

Specific points

In Fig 1, the authors mention that are working with 50 different surgical specimens, however, for the analysis and as shown in Fig 1d only 48 samples were shown. Hence, it is not clear why the other two samples were excluded for further analysis. Further, in supplemental material it was mentioned that only samples with more than 50 counts with respect to housekeeping gene expression were included. However, in Fig 1d sample P1 had only 50 housekeeping genes that according to the selection criteria should have been excluded from further analysis due to low quality. Also, in Fig 1f, sample P3 was shown as an example with high CD8 staining, however, sample P3 also appeared to be the most mutated of all the specimens, which might not be best representative sample to be shown. Perhaps the authors could show examples of another intermediate CD8 expressing samples. Fig 1d in the figure is wrongly annotated as Fig 1e and should be corrected.

In Fig 2, authors need to specify the selection criteria for the samples chosen for scRNA-seq analysis. Along the text, we suggest to use term "cytotoxic lymphocytes" instead of "cytotoxic cells". Likewise, in the text it is said "... and cytotoxic T cells (including CD8 T cells co-clustered with NK cells)", while we suggest to distinguish NK from cytotoxic CD8+ T cells in the figures and throughout the manuscript. In Fig 2c, it should be specified what kind of leukocytes (myeloid vs lymphocytic cells) are described and analyzed. Also, in the text it is mentioned that TAMs are actively proliferating and it can be seen in Fig 2c, which needs to be specified likewise. There is an error in the legend of figure 2, Fig 2b is shown as c and vice versa.

In Fig 3, authors need to specify the selection criteria for the samples chosen (cf. ext. data 5) and why the samples differ when comparing to the samples showing in Fig 2 (cf. ext. data 4).

Furthermore, out of those to why have the P6 and P9 been chosen to do a longitudinal study. In figure 3f the graph in the left is n=8 and the one in the right is n=6, which samples have been chosen? In the beginning of the legend for figure 3 it is written aA) when it should be a). Also, it should be commented on the time point of CSF taking with respect to the surgical resection of the tumors, as this potentially could have a strong impact on the CSF cellular compartment.

In Fig 4, the phrasing referring to cytotoxic lymphocytes is confusing. In Fig 4a-c, when cytotoxic cells are mentioned, they always include NK cells, however, at 4d it seems like that NK cells are not included. The authors need to clarify the selection criteria or correct the annotation. In the legend section f and g miss annotation letters.

In the methods section "cell sorting for single-cell RNA sequencing" it is mentioned that a total of 19 samples was collected, whereas in the main text only 16 samples were included in the analysis. In the first paragraph of the supplementary data on page 4 when mentioning "Extended data i, m" it should be mentioned that this data belongs to Fig 4.

In extended data 4, authors need to specify the selection criteria for the samples chosen (cf. Fig 3). In Fig 4f, the methanol preservation definitely looks inferior even after batch correction as compared to freshly processed or cryopreserved samples. Hence, I would suggest to perform and show separate analyses for each sampling condition, as cryoprotection and methanol might have substantial impacts on RNA integrity. Therefore, it would be interesting to see the RNA integrity numbers for the samples not freshly processed. Demonstrating that there are no major differences

between the different sampling conditions would be important and helpful for subsequent studies when choosing the optimal processing strategy. Also, it should be specified which samples belong to which sampling group.

In extended data 5 Fig 5a, the authors entitle in the x axis 3 time points as t1, t2 and t3, whereas in the corresponding main Fig 3 they use t0, t1, t2. This needs to be corrected and revised.

Reviewer #2 (Remarks to the Author):

The authors compared immune cells infiltrating brain metastasis with their counterparts in the CSF and report a level of similarity that allows monitoring tumor treatment response based on CSF immune cell analysis. The design of the study is very advanced and comprehensive; the data look solid and are presented in a clear, albeit sometimes over-compact, manner.

In spite of the evidence that is provided, I wonder how conclusive the data is that the phenotype of cytotoxic cells in the CSF recapitulates that in the tumor. In the healthy brain, the parenchyma is populated mostly by CD8+ T cells, while CD4+ T cells dominate in the CSF. Moreover, parenchymal T cells bear a Trm-cell profile (Smolders et al, Nat Commun 2018), while cells from CSF more resemble Tcm cells. The supplementary data seem to confirm the difference in CD4/CD8 ratio (Ext. data 5i) and may indicate a Trm phenotype of the tumor T cells (Ext. data 4i). However, the paper seems to ignore these findings and CD4+ T cells in general. Rather, the authors give the impression that cytotoxic T and NK cells, as a whole, are very similar between tumor and CSF. To further clarify this, data on CD8/CD4 ratio and a comparison of the phenotype of CD8+ and CD4+ T cells in both compartments should be provided. Even if only a small fraction of CSF T cells mimics their tumor counterparts, this may allow monitoring anti-tumor immunity. The broad similarity that now is suggested is not sufficiently supported in my view and even may be unlikely based on what we know about brain T cells.

Additional comments

1. Throughout the entire manuscript, information is provided in a very condensed way, and it requires a close look into the extended information to understand the data. I'm not opting for reshuffling the displayed items but feel that readability would benefit from a little more explanation.
2. I'm confused by the use of the term (predictive) biomarker. In my view, a biomarker mostly is a particular molecule. The authors use the term wider, eg for TME, IFN γ signature or T-cell tumor infiltration, which sounds odd to me. I suggest to simply remove 'as a biomarker' in line 3 of the abstract and 'are biomarkers used to' in line 10/11 of the introduction.

Reviewer #3 (Remarks to the Author):

The manuscript by Rubio-Perez et al describes the analyses of cerebrospinal fluid (CSF) as a mode of non-invasive method for selection of brain metastases patients with inflammatory signatures and stratify them for immune checkpoint inhibition (ICI) therapies. Using a combination of approaches including scRNA-seq and TCR sequencing they demonstrate that the signatures between tumors and the CSF are similar. In a few cases where longitudinal monitoring was possible, CSF analyses revealed clonal expansion of a subset of T cells. Overall, this is an exciting study that opens up new modes of immune monitoring in patients with brain metastases.

However, the authors have to address the following concerns:

- 1) The authors state that CD45+ cells were enriched in CSF prior to sequencing. It is important to show the flow cytometry scatter plots in the supplementary data. Gating strategies should be shown with isotype controls. Also, were the authors able to distinguish the CD45^{high} and CD45^{low} sub-populations which mark brain resident and infiltrating immune cells?
- 2) It is not clear how many samples had matched scRNA-seq AND flow cytometry data
- 3) The authors must show IFN γ expanded 10 gene signature that the authors used if possible. This will give an idea about the variation of the gene expression patterns within the IFN γ gene module.
- 4) How were the 9 patients for scRNA-seq selected? Is it based on availability of tissue?

5) A general rule of thumb is to use at least 3000-4000 cells per sample in scRNAseq. The authors have used less than 2000. Is this due to viability issues or technical factors? The authors must at least provide a justification.

6) Another major concern with respect to the scRNA-seq data in this study is there no detailed description of how the authors addressed a) batch effects, b) doublet removal, c) cell debris or low complexity library issues, d) RBC contamination, d) low quality cells that usually express mitochondrial genes. These are very important to address before data analyses can be performed.

7) The t-SNE plots showing the different clusters must also show markers that define sub-populations of myeloid and lymphoid lineage. Furthermore, in contrast to recent publications (Fiebel etl., Klemm et al., Cell 2020), the authors have equal proportions of microglia and macrophages, which is not the case according to both studies. The Cell papers show that microglia numbers diminish in brain metastases compared to TAMs.

8) Have the authors collected peripheral blood (PBMCs) from the same cohort of patients? It seems to me that the authors missed an important opportunity to compare the immune cell landscape between peripheral blood and those in the CSF. If the collections have been made, at minimum a multi-color flow comparison of cell type differences in the three compartments (tumor, CSF and PBMCs) should be made.

9) The discussion is very brief and lacks insight. The authors must expand this sections to place their study in the backdrop of recent studies by Johanna Joyce and others (Cell, 2020). Also, how does the CSF immune profiles from this study compare to other similarly inflamed brain disorders? For example, the authors could compare the profiles from this study to Schafflick et al Nat. Commun, 2020.

REVIEWER

COMMENTS

Reviewer #1 (Remarks to the Author):

This is an interesting study by Rubio-Perez and colleagues about studying cell-type specific diversity and transcriptomic changes in the CSF of patients with brain metastasis tumor formation. The group addressed the question whether the cellular/molecular diversity in the CSF does reflect changes in the actual tumor microenvironment. They applied scRNA-seq in combination with TCR genotyping, and found conserved TCR clonotypes between matched CSF and brain tumor samples. Also, by characterizing the immune cell signatures in the CSF, predictions about responses towards immune checkpoint inhibition were made. In summary, this manuscript follows-up on the idea that by analyzing a "liquid biopsy" compartment such as the CSF, it might be possible to study the microenvironment of solid tumors and predict response patterns to specific immune cell therapies. Overall, this is a well conducted and elegant study, however, several points remain that need to be addressed. Also, it would be interesting to see if further differences between the variable primary tumors can be further demonstrated with respect to immune subtypes and TCR subclones, which would be critical when tailoring immune therapies.

We thank the reviewer for considering our work "interesting", "well conducted" and "elegant study". We also agree with the reviewer that it would be interesting to observe further differences between primary tumors. However and unfortunately, the number of samples does not allow us to study those differences. Larger future studies are warranted to fully understand the potential differences in immune cell subtypes and TCR subclones among diverse tumor types.

Specific points

1) In Fig 1, the authors mention that are working with 50 different surgical specimens, however, for the analysis and as shown in Fig 1d only 48 samples were shown. Hence, it is no clear why the other two samples were excluded for further analysis.

We thank the reviewer for his/her comments and apologize for the lack of clarity. Due to sample availability not all samples underwent all the experimental analysis. There were no selection criteria but just limitation of sample availability and, in addition, as the reviewer commented, some samples were discarded due to quality thresholds.

In Fig 1d, only 48 samples were analyzed because we did not have enough tissue to perform IHC and Nanostring analysis in all 50 samples.

Further, in supplemental material it was mentioned that only samples with more than 50 counts with respect to housekeeping gene expression were included. However, in Fig 1d sample P1 had only 50 housekeeping genes that according to the selection criteria should have been excluded from further analysis due to low quality.

In some cases, we had sufficient material to perform duplicate analysis of the same sample. These sample duplicates are labeled as Px_2 in Ext. data 2c. In the particular case of P1, P1_2 was above the quality threshold and included in the analysis but not the P1 sample. When two

samples of the same case were processed, the one with the highest geometric mean of housekeeping genes was kept. This is now specified in the Methods section. We have also included this information in Ext. data 2 figure legend and illustrated in Ext. data 2c.

In order to clarify how samples were used across the experimental techniques, we have generated a Venn-diagram illustration and a scheme (new Ext. Data 2a, 2b) to show the sample availability. We have also modified Figure 1a in order to clarify in which samples TCRseq was performed. Moreover, Figures 2, 3 and 4 have now schemes representing the sample cohort being analyzed in each Figure. We have also modified the methods section to clarify the distribution of the samples across the experimental procedures.

2) Also, in Fig 1f, sample P3 was shown as an example with high CD8 staining, however, sample P3 also appeared to be the most mutated of all the specimens, which might not be best representative sample to be shown. Perhaps the authors could show examples of another intermediate CD8 expressing samples.

We thank the reviewer for his/her comments and we agree with him/her. We are now showing another more representative sample (sample P34) instead of P3. In addition, following the reviewer's request we have now included another intermediate CD8 expressing sample (sample P47). The new Fig 1f shows now representative IHC images of low, intermediate and high CD8⁺ T cell infiltration.

3) Fig 1d in the figure is wrongly annotated as Fig 1e and should be corrected.

We apologize for this mistake. This is now corrected.

4) In Fig 2, authors need to specify the selection criteria for the samples chosen for scRNA-seq analysis.

We thank the reviewer for the comment. The selection criterion was just sample availability. In some cases, we did not obtain a CSF sample from the patient or did not have enough tumor sample to perform the isolation of CD45⁺ cells and hence scRNA-seq. This will be stated in the text.

Along the text, we suggest to use term "cytotoxic lymphocytes" instead of "cytotoxic cells".

We agree with the reviewer and we are now using the term "cytotoxic lymphocytes".

Likewise, in the text it is said "... and cytotoxic T cells (including CD8 T cells co-clustered with NK cells)", while we suggest to distinguish NK from cytotoxic CD8⁺ T cells in the figures and throughout the manuscript.

We thank the reviewer for raising this point. In fact, we already tried to split the cytotoxic lymphocyte cluster into cytotoxic CD8⁺ T cells and NKs. However, due to the low number of sequenced cells belonging to this cell cluster and the large amount of shared markers between cytotoxic CD8⁺ T and NK cell types (i.e. cytotoxic cytokines), the cluster was not well separated into CD8⁺ T and NK cell types. Still, we observed that some of the cell clusters tended to be more enriched in T than NK cells and that is why we labeled them differently: Cytotoxic lymphocytes [Tcell + NK], Cytotoxic lymphocytes [>> Tcell], Cytotoxic lymphocytes [>> NK].

Please, see below the violin and scatter plots representing the expression levels of the main markers of cytotoxic T and NK cells across the clusters of cytotoxic lymphocytes associated with Figure 2. For this analysis, Naïve T cells and Tregs were excluded. Note that NKG7 and GNLY (markers of NK cells) are expressed across all cell clusters.

The three cytotoxic lymphocyte clusters were identified using a granularity of 0.3, the same granularity used to identify the proliferating leukocyte clusters. We tried to increase the granularity to 0.4 and we obtained 6 distinct clusters instead of three. Please, see the UMAP representation below:

However, we observed that the same markers analyzed before were present in all cell clusters. Please, see violin plots below.

Note that even if we were able to identify an NK enriched cluster (cluster #4), we found NK markers in the remaining clusters. Therefore, we decided not to increase the granularity, as the newly identified cell clusters had few cells and could be considered biologically not meaningful.

In Fig 2c, it should be specified what kind of leukocytes (myeloid vs lymphocytic cells) are described and analyzed. Also, in the text it is mentioned that TAMs are actively proliferating and it can be seen in Fig 2c, which needs to be specified likewise.

We thank the reviewer for the comment and we apologize for the lack of clarity. We have repeated the analysis and now we have better identified the proliferating cell clusters. This is shown in the new Fig 2c.

There is an error in the legend of figure 2, Fig 2b is shown as c and vice versa.

We thank the reviewer for spotting this mistake. We have now corrected it in the new figure legend.

In Fig 3, authors need to specify the selection criteria for the samples chosen (cf. ext. data 5) and why the samples differ when comparing to the samples showing in Fig 2 (cf. ext. data 4).

In Figure 3, we only included tumor-CSF matched samples, while Figure 2 includes all tumor samples. This has now been clarified by adding a scheme of the sample cohort used in Figure 2a, 3a, and by specifying the sample cohort in the figure legend as well as in Ext. data 2b.

Furthermore, out of those to why have the P6 and P9 been chosen to do a longitudinal study.

Due to the status of the disease, we could not obtain longitudinal samples of CSF from all patients. We did not select the samples to be analyzed but we performed the analysis in the cases where we had longitudinal samples of CSF available.

In figure 3f the graph in the left is n=8 and the one in the right is n=6, which samples have been chosen?

We apologize for the lack of clarity. We have now modified Fig 3a and its figure legend to clarify the two cohorts of samples used in the graphs. In the left panel, we performed flow cytometry in 16 matched tumor-CSF samples from 8 patients, and in the right panel, we performed scRNA-seq of 16 samples of 6 patients where we had matched tumor-CSF samples. There are thus two cohorts of samples that are now shown and described in the figure and figure legend.

In the beginning of the legend for figure 3 it is written aA) when it should be a).

We want to thank the reviewer again for his/her thorough revision that is greatly improving our manuscript. We have now corrected the error.

Also, it should be commented on the time point of CSF taking with respect to the surgical resection of the tumors, as this potentially could have a strong impact on the CSF cellular compartment.

We apologize for the lack of clarity. If not specified all CSF samples were collected at the time of surgery. We have now rewritten the figure legend to specify the time of CSF collection.

In Fig 4, the phrasing referring to cytotoxic lymphocytes is confusing. In Fig 4a-c, when cytotoxic cells are mentioned, they always include NK cells, however, at 4d it seems like that NK cells are not included. The authors need to clarify the selection criteria or correct the annotation.

In Figure 4, only TCR+ cells have been included. Therefore, all cells shown are T cells. However, for the sake of consistency with previous figures the same cell cluster nomenclature was used. In order to clarify which samples were considered, we have generated a new panel b) illustrating the sample cohort analyzed, and we have changed accordingly the figure legend and Ext. data 2b.

In the legend section f and g miss annotation letters.

We thank the reviewer for spotting this error. We have now corrected it in the figure legend.

In the methods section "cell sorting for single-cell RNA sequencing" it is mentioned that a total of 19 samples was collected, whereas in the main text only 16 samples were included in the analysis.

Figure 4 develops from Figure 3 where only matched tumor-CSF samples were included. Therefore a total of 16 samples were considered: 12 matched tumor-CSF samples at time 0 and four additional CSF samples from later time points. On the other hand, a total of 19 samples were profiled by scRNAseq, the aforementioned 16 samples plus 3 samples where only the tumor could be analyzed. As explained previously, we have now clarified this in Ext. data 2 a, b and additional schemes in the main figures.

In the first paragraph of the supplementary data on page 4 when mentioning “Extended data i, m” it should be mentioned that this data belongs to Fig 4.

We thank the reviewer for spotting this error. We have now corrected it in the figure legend.

In extended data 4, authors need to specify the selection criteria for the samples chosen (cf. Fig 3).

We thank the reviewer for raising this point. We have now modified the figure legend in order to specify the selection criteria.

In Fig 4f, the methanol preservation definitely looks inferior even after batch correction as compared to freshly processed or cryopreserved samples. Hence, I would suggest to perform and show separate analyses for each sampling condition, as cryoprotection and methanol might have substantial impacts on RNA integrity. Therefore, it would be interesting to see the RNA integrity numbers for the samples not freshly processed. Demonstrating that there are no major differences between the different sampling conditions would be important and helpful for subsequent studies when choosing the optimal processing strategy. Also, it should be specified which samples belong to which sampling group.

We apologize for the lack of clarity related to the methanol batch correction. We have now labeled all the patient samples in Ext. data 4b,c and Ext. data 5a,c according to the preservation technique used in each case. As it can be observed in Ext. data 4b,c; the number of cells recovered after methanol preservation were lower compared to fresh samples (Ext. data 4b). However, the RNA quality of methanol preserved samples (measured as the number of expressed genes per cell) is comparable to the one of non-methanol preserved samples (Ext. data 4c); suggesting that both sample types are comparable. The same observations can be made in the CSF samples in Ext. data 5a,c.

A thorough analysis comparing methanol-preserved and fresh samples was carried out in tumor samples. In Ext. data 4f, we show that a batch effect by sample preservation technique can be corrected by applying the sample “*integration*” method of the Seurat pipeline. To better illustrate this point, we have included a box plot in Ext. data 4f which illustrates the change of entropy across cell clusters by the preservation technique before and after batch correction. It can be observed how entropy significantly increases (i.e. cells are more equally distributed across sample preservation techniques per cell cluster rather than all cells in a cell cluster belonging to the same sample preservation technique) after batch normalization. We have now better discussed these points in the text of the manuscript and in the figure legend of Ext. data 4, and we have properly updated the codes in the GitHub repository to allow the reproduction of this new analysis.

In extended data 5 Fig 5a, the authors entitle in the x axis 3 time points as t1, t2 and t3, whereas in the corresponding main Fig 3 they use t0, t1, t2. This needs to be corrected and revised.

We thank the reviewer for spotting this error. We have now corrected it.

Reviewer #2 (Remarks to the Author):

The authors compared immune cells infiltrating brain metastasis with their counterparts in the CSF and report a level of similarity that allows monitoring tumor treatment response based on CSF immune cell analysis. The design of the study is very advanced and comprehensive; the data look solid and are presented in a clear, albeit sometimes over-compact, manner.

We thank the reviewer for considering our study “very advanced and comprehensive” and that the data “look solid and are presented in a clear, albeit sometimes over-compact, manner”. In terms of the text being compact, we have now expanded the description and explanation of the experiments in order to clarify the results.

In spite of the evidence that is provided, I wonder how conclusive the data is that the phenotype of cytotoxic cells in the CSF recapitulates that in the tumor. In the healthy brain, the parenchyma is populated mostly by CD8+ T cells, while CD4+ T cells dominate in the CSF. Moreover, parenchymal T cells bear a Trm-cell profile (Smolders et al, Nat Commun 2018), while cells from CSF more resemble Tcm cells. The supplementary data seem to confirm the difference in CD4/CD8 ratio (Ext. data 5i) and may indicate a Trm phenotype of the tumor T cells (Ext. data 4l). However, the paper seems to ignore these findings and CD4+ T cells in general. Rather, the authors give the impression that cytotoxic T and NK cells, as a whole, are very similar between tumor and CSF. To further clarify this, data on CD8/CD4 ratio and a comparison of the phenotype of CD8+ and CD4+ T cells in both compartments should be provided. Even if only a small fraction of CSF T cells mimics their tumor counterparts, this may allow monitoring anti-tumor immunity. The broad similarity that now is suggested is not sufficiently supported in my view and even may be unlikely based on what we know about brain T cells.

We thank the reviewer for these comments. The objective of our work was not to perform a detailed characterization of the immune cells in the CSF, but focus on evaluating the levels of cytotoxic lymphocytes since they are relevant to determine ICI clinical responses. Future studies are warranted to thoroughly characterize in detail each of the immune cell types present in the CSF to be able to compare them with the ones present in the tumor lesions. That is why, unfortunately, we did not fully characterize the CD4⁺ T cells or subclassify them into Trm or Tcm.

Still and following the reviewer’s suggestion, we have further analyzed the levels of CD4⁺ T cells in the tumor and CSF samples. We have now generated a new panel, Ext. data 4o to better illustrate the CD4 expression levels across scRNAseq cell clusters. As it can be observed, the levels of CD4 expression across T cell clusters from scRNAseq are low; being the Treg cell cluster the one with the higher % of cells expressing CD4 (17.5% of the cells expressing CD4). Because of this low amount of cells expressing CD4, we did not feel confident enough to increase cell cluster granularity and define a CD4⁺ T cell group.

We would like to clarify that Ext. data 5i (now 5j) is only an example of the gating strategy representing a single sample. We have now added CD4⁺ data from flow cytometry. CD4⁺ T cell gate strategy has also now been included in Ex. data 5j. We have now included the CD4⁺ T cell percentage (the total percentage and the relative percentage to CD45⁺ cells) from tumor and CSF samples in Ext. data 1f. Consequently, we have expanded Figure 3e, f flow cytometry

analyses including the results for all T cells identified (adding CD4⁺ T cells). We have explained these results in the main manuscript.

Due to the reasons stated above regarding CD4 expression in scRNA-seq data, we have only evaluated the CD4⁺/CD8⁺ T cell ratio using flow cytometry. This new analysis has been included as Ext. data 5m. We observe a significantly higher CD8/CD4 ratio (Wilcoxon test p-value = 0.01) in the tumor vs the CSF. This observation supports the reviewer comments where CD8⁺ T cells are enriched in the tumor and CD4⁺ T cells in the CSF.

We have now discussed this new analysis in the main manuscript.

All new analyses have been included in the GitHub code repository and are fully reproducible.

Additional comments

1. Throughout the entire manuscript, information is provided in a very condensed way, and it requires a close look into the extended information to understand the data. I'm not opting for reshuffling the displayed items but feel that readability would benefit from a little more explanation.

We thank the reviewer for this comment. We have now expanded the description and explanation of the experiments in order to clarify the results.

2. I'm confused by the use of the term (predictive) biomarker. In my view, a biomarker mostly is a particular molecule. The authors use the term wider, eg for TME, IFN γ signature or T-cell tumor infiltration, which sounds odd to me. I suggest to simply remove 'as a biomarker' in line 3 of the abstract and 'are biomarkers used to' in line 10/11 of the introduction.

We have followed the reviewer suggestion and removed the word biomarker from the mentioned text.

Reviewer #3 (Remarks to the Author):

The manuscript by Rubio-Perez et al describes the analyses of cerebrospinal fluid (CSF) as a mode of non-invasive method for selection of brain metastases patients with inflammatory signatures and stratify them for immune checkpoint inhibition (ICI) therapies. Using a combination of approaches including scRNA-seq and TCR sequencing they demonstrate that the signatures between tumors and the CSF are similar. In a few cases where longitudinal monitoring was possible, CSF analyses revealed clonal expansion of a subset of T cells. Overall, this is an exciting study that opens up new modes of immune monitoring in patients with brain metastases. However, the authors have to address the following concerns:

We thank the reviewer for considering our study "an exciting study that opens up new modes of immune monitoring in patients with brain metastases".

1) The authors state that CD45+ cells were enriched in CSF prior to sequencing. It is important to show the flow cytometry scatter plots in the supplementary data. Gating strategies should

be shown with isotype controls. Also, were the authors able to distinguish the CD45^{high} and CD45^{low} sub-populations which mark brain resident and infiltrating immune cells?

We thank the reviewer for his/her comments and apologize for the lack of clarity. CSF cells were collected by centrifuging the CSF sample and no CD45⁺ enrichment was performed. This is now clarified and moved to the main Methods section.

We are now showing the flow cytometry scatter plots in the supplementary data and we have included the CD45⁺ gating strategy showing isotype controls of the two different CD45 antibodies that were used in the flow cytometry panels. In total, two new supplementary figures have been generated (Ext. data 5i, Ext. data 5k). In Ext. data 5i, blank (without antibodies) was compared to samples stained with all the antibodies plus CD45 isotype control (IgG1k - BV510) (labeled as isotype control) or anti-human CD45 BV510 (labeled as anti-CD45). These data correspond to the panel which has been used to determine immune cell populations, as shown in Ext. data 5j. In Ext. data 5k, blank (without antibodies) was compared to samples stained with all the antibodies plus CD45 isotype control (IgG1k- PeCy5) (isotype control) or anti-human CD45-PeCy5 (anti-CD45). These data correspond to the panel which has been used to determine T cell subpopulations, as shown in Ext. data 5l.

In both figures (Ext. data 5i, and Ext data 5k) a representative sample of PBMCs (control sample), CD45⁺ isolated cells from tumor samples, and CSF cells are displayed as indicated.

We went through our data to analyze the CD45 high vs low populations and although we observed these populations in some samples, some others just presented a continuum of CD45 expression. Unfortunately, due to the limited number of samples we were unable to draw statistical conclusions from our results. For this reason, we decided to consider CD45 as a unique population.

2) It is not clear how many samples had matched scRNA-seq AND flow cytometry data

We thank the reviewer for this comment that is similar to reviewer #1's comment. As we answered reviewer #1, we have now generated a new Ext. data 2a containing a Venn diagram illustration of patient sample availability across the distinct experimental techniques; and a new Ext. data 2b illustrating the patient samples being analyzed with different techniques. In Ext. data 2a, we now show that 7 tumor samples were processed by both scRNA-seq and flow cytometry. This has now also been specified in the Methods section.

3) The authors must show IFN γ expanded 10 gene signature that the authors used if possible. This will give an idea about the variation of the gene expression patterns within the IFN γ gene module.

We thank the reviewer for the suggestion. We have now included a heatmap representing the gene expression of the IFN γ signature in Ext. data 2g. Please, note that we have used the refined signature (Table 2, Ayers et. al. 2017) for which statistical clinical associations were shown (n=6 genes: IDO1, CXCL10, CXCL9, HLA-DRA, STAT1, IFNG). Thus, a heatmap with the expression of these six genes across samples is shown. To provide more insights into the gene expression variability, a dendrogram at the gene level has been also included in Ext. data 2g.

In addition, we have now included the list of the gene sets used to compute the enrichment scores in the GitHub repository.

4) How were the 9 patients for scRNA-seq selected? Is it based on availability of tissue?

Indeed, no selection criterion was undertaken other than tissue availability.

5) A general rule of thumb is to use at least 3000-4000 cells per sample in scRNA-seq. The authors have used less than 2000. Is this due to viability issues or technical factors? The authors must at least provide a justification.

We thank the reviewer for this comment. Indeed, sometimes we used less than 2000 due to low amount of cells present in the CSF, technical or viability issues.

6) Another major concern with respect to the scRNA-seq data in this study is there no detailed description of how the authors addressed a) batch effects, b) doublet removal, c) cell debris or low complexity library issues, d) RBC contamination, d) low quality cells that usually express mitochondrial genes. These are very important to address before data analyses can be performed.

We thank the reviewer for raising this point and we apologize for not describing these points in the previous version of the manuscript. We have now included a detailed answer to all the points in the Methods section of the manuscript.

a) **batch effects:** We have expanded the visualization of the batch adjustment by adding an additional analysis in Ext. data 4f and adding sample preservation technique annotation in Ext. data 4b,c and Ext. data 5a,c. As it can be observed in Ext. data 4b,c, while the number of cells recovered after methanol preservation were lower (Ext. data 4b) compared to fresh samples, the RNA quality of the methanol preserved samples (measured as the number of expressed genes per cell) is comparable to the one of non-methanol preserved samples (Ext. data 4c). The same observations can be made in the CSF samples in Ext. data 5a,c. A thorough analysis of the batch effect was carried out in tumoral samples. In Ext. data 4f we demonstrate that a batch effect by sample preservation technique can be corrected by applying the sample *"integration"* method of the Seurat pipeline. To better illustrate this, we have included a box plot in Ext. data 4f which illustrates the change of entropy across cell clusters by preservation technique before and after batch correction. It can be observed how entropy significantly increases (i.e. cells are more equally distributed across sample preservation techniques per cell cluster rather than all cells in a cell cluster belonging to the same sample preservation technique) after batch normalization. We have now better discussed this in the figure legend of Ext. data 4 and we have properly updated the codes in the GitHub repository to allow the reproduction of this new analysis.

b) **doublet removal:** We excluded cells with an elevated gene count (>2500 genes) as potential doublets. Doublets were further identified through manual inspection of clusters with mixed phenotypes, assessed through canonical cell type markers (e.g. CD3E/CD3D for T-cells and CD68/CD163 for myeloid cells).

c) cell debris or low complexity library issues, d) RBC contamination, d) low quality cells that usually express mitochondrial genes: Cells with <100 genes (tumors) or <70 genes (CSF) detected were considered as low quality and were excluded from the analysis. Likewise, cells with high mitochondrial gene content (>10% for tumors and >20% for CSF) were considered as damaged and removed from the analysis. These filters also excluded contaminating erythrocytes and droplet with cell debris, which presented extremely low gene counts. Successful removal of erythrocytes was further confirmed by the inspection of canonical cell type markers, such as hemoglobin genes.

7) The t-SNE plots showing the different clusters must also show markers that define sub-populations of myeloid and lymphoid lineage.

We thank the reviewer for raising this point and we agree with the reviewer that this visualization needs to be included. We have generated new figure panels (Ext. data 4h, m and q) including the t-SNE plots. In addition, the markers that define the cell populations can be evaluated in depth through the dot plots in Ext. data 4i, k, n, r which represent the percentage of cells expressing each marker per cell cluster, as well as the average expression of the gene. Moreover, the list of all markers defining the distinct immune cell populations is available in the GitHub repository.

Furthermore, in contrast to recent publications (Fiebel et al., Klemm et al., Cell 2020), the authors have equal proportions of microglia and macrophages, which is not the case according to both studies. The Cell papers show that microglia numbers diminish in brain metastases compared to TAMs.

We have compared the markers we have used to define microglia and TAMs with the ones used by Fiebel et al. 2020 and Klemm et al. 2020. Of note, both studies mainly rely on the markers described in Bowman et al 2016. To give more strength to the TAM vs microglia cell cluster annotations, and following Klemm et al. 2020 rationale; we have downloaded the markers of macrophage and microglia from Bowman et al. 2016 and performed a Gene Set Enrichment Analysis (GSEA) on the list of ranked genes by fold-change of the differential expression analysis between TAM and microglia. Accordingly, we found the macrophage gene set up-regulated and the microglia gene set significantly down-regulated (this analysis has now been included as Ext. data 4k, right panel, and further discussed in the figure legend).

Thus, we confirmed that our study could differentiate TAM and microglia. However, our results did not show differences in the relative abundance of TAM and microglia as shown by Fiebel et al. 2020 and Klemm et al. 2020. This could be due to the relatively small cohort and tumor heterogeneity of our study. Larger cohorts need to be analyzed to statistically address this point.

This new analysis has been explained in the Supplementary Methods section and we have properly updated the codes in the GitHub repository to allow the reproduction of it.

8) Have the authors collected peripheral blood (PBMCs) from the same cohort of patients? It seems to me that the authors missed an important opportunity to compare the immune cell

landscape between peripheral blood and those in the CSF. If the collections have been made, at minimum a multi-color flow comparison of cell type differences in the three compartments (tumor, CSF and PBMCs) should be made.

We fully agree and very unfortunately we do not have access to the PBMCs from the same cohort of patients. However, we believe that this is a great point and we are planning to engage a prospective study to address this question.

9) The discussion is very brief and lacks insight. The authors must expand this sections to place their study in the backdrop of recent studies by Johanna Joyce and others (Cell, 2020). Also, how does the CSF immune profiles from this study compare to other similarly inflamed brain disorders? For example, the authors could compare the profiles from this study to Schafflick et al Nat. Commun, 2020.

We agree with the reviewer and we have now expanded the discussion section commenting the Cell papers and the Schafflick et al. work.

REVIEWERS' COMMENTS

Reviewer #1 (Remarks to the Author):

The authors have replied to all relevant points that were raised. There are no further comments.

Reviewer #2 (Remarks to the Author):

My comments have been adequately addressed. Jörg Hamann

Reviewer #3 (Remarks to the Author):

The manuscript by Rubio-Perez et al report the utility of immune infiltrates of the CSF for non-invasive profiling. They perform scRNA-seq of CSF infiltrates and demonstrate a match between tumor and CSF immune profiles. Furthermore the T cell clones were similarly expanded between brain and CSF. Overall the study opens up a very exciting avenue of liquid biopsy for monitoring responses to immunotherapy in brain metastases. The authors have now responded appropriately to all the critiques raised in the first review and the manuscript is now acceptable for publication.

Signed,
Krishna Bhat